

# MHC class I allele diversity in cynomolgus macaques of Vietnamese origin

Shuting Huang[*], Xia Huang[*], Shuang Li, Mingjun Zhu and Min Zhuo

School of Biology and Biological Engineering, South China University of Technology, Guangzhou, Guangdong, China

[*] These authors contributed equally to this work.

## ABSTRACT

Cynomolgus macaques (*Macaca fascicularis*, *Mafa*) have been used as important experimental animal models for carrying out biomedical researches. The results of biomedical experiments strongly depend on the immunogenetic background of animals, especially on the diversity of major histocompatibility complex (MHC) alleles. However, there is much less information available on the polymorphism of MHC class I genes in cynomolgus macaques, than is currently available for humans. In this study, we have identified 40 *Mafa-A* and 60 *Mafa-B* exons 2 and 3 sequences from 30 unrelated cynomolgus macaques of Vietnamese origin. Among these alleles, 28 are novel. As for the remaining 72 known alleles, 15 alleles are shared with other cynomolgus macaque populations and 32 are identical to alleles previously reported in other macaque species. A potential recombination event was observed between *Mafa-A1*091:02* and *Mafa-A1*057:01*. In addition, the *Mafa-A1* genes were found to be more diverse than human *HLA-A* and the functional residues for peptide binding sites (PBS) or TCR binding sites (TBS) in *Mafa-A1* have greater variability than that for non-PBS or non-TBS regions. Overall, this study provides important information on the diversity of *Mafa-A* and *Mafa-B* alleles from Vietnamese origin, which may help researchers to choose the most appropriate animals for their studies.

# INTRODUCTION

The MHC glycoproteins, usually known as MHC class I and class II molecules, play important roles in the regulation of innate and adaptive immune response. The MHC classical class I molecules contribute both to innate immunity, by engaging Natural Killer (NK) cell receptors, and to adaptive immunity, by presenting antigens to CD8[+] T cells to induce their activation and cytotoxicity (*Parham, 2005*). Correlating with these functions, the antigen-binding sites, which are mostly located throughout the $\alpha$1 and $\alpha$2 domains encoded by the exons 2 and 3, exhibit the highest polymorphism within the full length of MHC class I gene sequences. Scholars found that the polymorphism of MHC genes is generated by a combination of mutation, recombination, and gene duplication and loss, and is maintained over time by selection (*Robinson et al., 2017*; *Steiner et al., 2002*; *Bahr &*

Corresponding author
Min Zhuo, zhuomin@scut.edu.cn

*Wilson, 2012*). Numerous infectious and autoimmune diseases are strongly associated with particular MHC alleles and haplotypes (*Nomura & Matano, 2012*; *Kuniholm et al., 2011*; *Shiina et al., 2017*; *Lenz et al., 2015*). For example, human susceptibility to rheumatoid arthritis (RA) was found linked strongly with certain MHC class I and II alleles, including *HLA-DRB1*, *HLA-DPB1* and *HLA-B* (*Raychaudhuri et al., 2012*). In response to HIV, human MHC class I genes, *HLA-B*57* and *HLA-B*27* exhibit strong and consistent association with lower viral loads in the chronic phase and slow disease progression (*Martin & Carrington, 2013*). In contrast, *HLA-B*35* and *HLA-B*58* associating with rapid disease progression have also been reported (*Nomura & Matano, 2012*). The cynomolgus macaque and the rhesus macaque (*Macaca mulatta*, *Mamu*), are both important nonhuman primate animal models for the study of various human diseases such as acquired immunodeficiency syndrome, tuberculosis, Alzheimer's disease, Parkinson's disease, diabetes, as well as transplantation researches and pharmacodynamic evaluation (*Vierboom et al., 2008*; *Smits et al., 2011*; *Lin et al., 2009*; *Kisu et al., 2014*; *Walter & Ansari, 2015*; *Wang et al., 2007*; *Emborg, 2007*). The rhesus monkey model of collagen-induced arthritis (CIA) is widely used to study the pathogenesis of human RA. It was reported that *Mamu-B*001* is resistant to CIA (*Bakker et al., 1992*). The CIA-susceptible rhesus monkeys need to be preselected on the basis of absence of *Mamu-B*001* (*Vierboom et al., 2016*). The Indian-origin rhesus macaque of SIV infection is firstly used as an AIDS model. *Mamu-A1*001*, *Mamu-A3*13:03*, *Mamu-B*008* and *Mamu-B*017* are known as protective alleles and macaques possessing these alleles tend to show slow disease progression after SIVmac251/SIVmac239 challenge (*Nomura & Matano, 2012*; *Walter & Ansari, 2015*; *Loffredo et al., 2007*; *Wambua et al., 2011*). Of note, several alleles, *Mamu-A1*001*, *Mamu-B*001*, and *Mamu-B*017* are distributed at high frequencies (*Allen et al., 1998*; *Mothe et al., 2002*). Additionally, anchor residues of CTL epitopes presented by *Mamu-B*017/Mamu-B*008* were indicated to be similar to those restricted by *HLA-B*57/HLA-B*27* (*Loffredo et al., 2009*). Due to the extensive characterization of several of these alleles, Indian rhesus macaque is the most widely utilized model in AIDS research. Since the export of rhesus macaques from India was restricted in 1978, the use of cynomolgus macaques for biomedical research has become increasingly prevalent. This species inhabits widely throughout Southeast Asia, including the Philippines, Indonesia, Vietnam, Malaysia, Thailand, Cambodia, and Brunei (*Gumert, 2011*). In addition, it also had been introduced to Mauritius island located in the western Indian Ocean about 400 years ago, where the Mauritius cynomolgus macaque had become an insular population (*Sussman & Tattersall, 1986*). Previous studies have demonstrated that most MHC class I alleles found in cynomolgus macaques are unique to animals from particular regions. The distribution frequencies of MHC alleles in distinct population are also different (*Otting et al., 2007*; *Pendley et al., 2008*; *Campbell et al., 2009*). For example, the MHC diversity of the Mauritius cynomolgus macaque is more limited than that of other populations (*Budde et al., 2010*). Hence, the information on MHC diversity from different regions as well as their association with various disease susceptibility needs to be considered carefully, when cynomolgus macaques were used in biomedical studies (*Seekatz et al., 2013*).

The human classical MHC class I genes, *HLA-A*, *HLA-B*, and *HLA-C*, exhibit high polymorphism. 5018 *HLA-A* and 6096 *HLA-B* alleles have been included in the IMGT/HLA database (Release 3.36.0, 2019-04-17), which is a module of The Immuno Polymorphism Database (IPD) (*Robinson et al., 2011*). In comparison, the orthologues of the *HLA-C* gene have not been identified so far in macaques and the *MHC-A* and *MHC-B* genes in macaques have more complex organization than the human genes (*Shiina et al., 1999*; *Shiina et al., 2015*; *Wiseman et al., 2013*). Only one copy of the *HLA-A* and *HLA-B* genes are present in humans, whilst seven A-like genes and up to nine B-like genes are present in macaques. Furthermore, a novel gene locus, *Mafa-A8*01:01*, was discovered recently in cynomolgus macaque of Filipino origin. Both the *MHC-A* and *MHC-B* loci are duplicated in cynomolgus macaque during evolution. Nevertheless, only 375 *Mamu-A* and 513 *Mamu-B* alleles, along with 494 *Mafa-A* and 717 *Mafa-B* alleles, have been deposited in the IPD-MHC database (Release 3.2.0.0.) (*Shiina et al., 2015*; *Maccari et al., 2016*). In comparison to their human counterparts, the number and the detailed information on the polymorphism of MHC classic class I genes in macaques are still lacking. The cynomolgus macaques bred in South China mainly originated from Vietnam and have been exported to various places in the world for biomedical research (*Karl et al., 2017*). To better understand the characteristics of their MHC class I alleles, the polymorphism analysis of both the *Mafa-A* and the *Mafa-B* exons 2 and 3 sequences were carried out simultaneously in 30 unrelated animals.

## MATERIAL AND METHODS

### Animals

All cynomolgus monkeys were housed in the South China Primate Research & Development Center (Guangdong, China) and were clinically asymptomatic for known diseases. Peripheral blood samples were collected from over 30 unrelated Vietnamese-origin cynomolgus macaques. The experiments were reviewed and approved by the Institutional Animal Care and Use Committee (IACUC) of Guangdong Landau Biotechnology Co. Ltd. (project number: IACUC-003).

### RNA extraction, cDNA cloning and sequencing of MHC class I genes

Total RNA was extracted from peripheral blood mononuclear cell samples of 30 animals using E.Z.N.A.™ Blood RNA Kits (OMEGA bio-tek). cDNA was synthesized using a PrimeScript™ II 1st Strand cDNA Synthesis Kit (TaKaRa Bio, Kusatsu, Japan). Amplification of the full length of exons 2 and 3 sequences was investigated using specific primer pairs (*Mafa-A*: 'A–F' 5′-AACCCTCCTCCTGGTGCTCT-3′, and 'A–R' 5′-GGAAGGTTCCATCTCCTGCAG-3′, *Mafa-B*: 'B–F' 5′-AACCCTCCTCCTGCTGCT-3′, and 'B–R' 5′-TGGACTGGGAAGATGGCT-3′) The two upstream primers are both located in exon 1 and the two downstream primers are located in exon 4. PCR employed a denaturation process for 3 min at 94 °C, followed by 32 cycles at 94 °C for 30 s, 58 °C (*Mafa-A*) or 56 °C (*Mafa-B*) for 30 s, 72 °C for 1 min, with a final process at 72 °C for 8 min. *Ex Taq* DNA polymerase (TaKaRa) was used in this reaction. PCR products were purified and cloned into the pMD18-T vector (TaKaRa). For each animal, about 30 clones were selected for *Mafa-A* and *Mafa-B* respectively and then were sequenced bidirectionally by

the service provider (Beijing Genomics Institute, Shenzhen, China). Nucleotide sequences of cDNAs were assembled and processed using SeqMan (DNASTAR, Madison, WI, USA *Burland, 2000*) and aligned using the Clustal W program (BioEdit (*Hall, 2015*)). To ensure authenticity, each sequence was uniquely named if three or more identical clones were observed from at least two individuals, or from two independent PCR for an individual. These sequences were then submitted to the GenBank for accession numbers and to the IPD–MHC database for allele nomenclature (*Robinson et al., 2013*; *De Groot et al., 2012*).

## Phylogenetic analysis

Recombination analysis was performed using the Recombination Detection Program version 4 (RDP4; *Martin et al., 2015*) with a window size of 20 nucleotides and *P* value less than 0.000005, and using the Recombination Identification Program (RIP) with a window size of 200 and 99% confidence intervals (http://www.hiv.lanl.gov/) (*Zhao et al., 2013*). A phylogenetic tree was constructed using the neighbor-joining (NJ) method (*Saitou & Nei, 1987*) in MEGA7 (*Kumar, Stecher & Tamura, 2016*) using exons 2 and 3. Evolutionary distances were computed using the Kimura 2-parameter model (*Kimura, 1981*) and assessed using 1,000 bootstrap replicates. Values greater than 50% were used as data-points to construct the tree. The nucleotide polymorphic sites were analyzed by DnaSP. The frequency for the second-most common nucleotide at each position was calculated by the number of occurrences of a nucleotide divided by the number of *Mafa-A1* sequences used in this analysis. The frequency for the second-most common amino acid at each position was also calculated by the number of occurrences of an amino acid divided by the number of *Mafa-A1* amino acid sequences used in this analysis (*Robinson et al., 2017*).

## RESULTS AND DISCUSSION

### Summary of the identified MHC class I alleles

cDNA clones were obtained by RT-PCR using *Mafa-A* and *Mafa-B* specific primer pairs. A total of 1965 clones were sequenced, and 882 *Mafa-A* and 859 *Mafa-B* cDNA sequences were acquired. After sequence alignment and filtering out the sequences detected identical in less than three clones, we identified 100 MHC class I alleles from 30 cynomolgus macaques of Vietnamese origin, including 40 *Mafa-A* and 60 *Mafa-B* genes, of which 28 alleles (11 *Mafa-A*, 17 *Mafa-B*) were identified as new ones. Their allele names, accession numbers, shared alleles in other cynomolgus macaque populations and counterparts in other macaque species are listed in Tables 1 and 2, respectively. Among the 28 novel alleles, five of them, namely *Mafa-A1*048:01* (KT907348), *Mafa-B*006:01:01* (KT895494), *Mafa-B*112:01* (KT895480), *Mafa-B*180:01* (KT895475) and *Mafa-B*202:01* (KT895441), were newly detected in cynomolgus macaque. The other 23 alleles are new at five- to seven-digit levels of classification. In recent years, it has been found that the epitranscriptomic modifications of mRNA play very important roles in the regulation of gene expression (*Peer, Rechavi & Dominissini, 2017*). However, any kind of epitranscriptomic modification on MHC RNAs hasn't been reported currently (*Grozhik & Jaffrey, 2017*). Therefore, we need to further study the existence of mRNA modifications in the transcripts of these 28 new alleles in the future. The remaining 72 alleles have been reported previously in the IPD-MHC database,

**Table 1    40 *Mafa-A* alleles detected in Vietnamese-origin cynomolgus macaques.**

| Allele name | Accession number | Other origin | Macaque counterparts[a] |
|---|---|---|---|
| *Mafa-A1*001:01:02* | KT907313 | *A1*001:01:01* (MacM)[c] | *Mamu-A1*001:01* (U50836-I) |
| *Mafa-A1*003:03* | KT907312 | | *Mamu-A1*003:01:01* (U41379-Unk) |
| *Mafa-A1*003:06:01* | KT907326 | | |
| **Mafa-A1*003:06:02** | KT907327 | | |
| *Mafa-A1*007:01* | KT907328 | | |
| **Mafa-A1*007:07** | KT907316 | | *Mamu-A1*007:02* (AF157397-Unk) |
| *Mafa-A1*015:01* | KT907351 | | *Mamu-A1*015:01* (AB551785-Bu) |
| *Mafa-A1*018:01* | KT907329 | | |
| **Mafa-A1*018:08** | KT907330 | | |
| *Mafa-A1*022:05* | KT907331 | | |
| *Mafa-A1*022:06* | KT907309 | | |
| *Mafa-A1*022:09:01* | KT907332 | | |
| *Mafa-A1*027:01* | KT907333 | | |
| *Mafa-A1*028:01* | KT907334 | | |
| *Mafa-A1*036:02* | KY073130 | | |
| *Mafa-A1*040:01:02* | KT907315 | | |
| *Mafa-A1*040:03* | KT907321 | | |
| **Mafa-A1*040:04** | KT907322 | | |
| *Mafa-A1*042:01* | KT907324 | | |
| *Mafa-A1*045:01* | KT907335 | | *Mamu-A1*045:01* (EU262741-Ch) |
| **Mafa-A1*048:01** | KT907348 | | |
| *Mafa-A1*053:01* | KT907336 | | *Mamu-A1*053:02* (EU551177-Ch) |
| *Mafa-A1*056:03:01* | KT907337 | | *Mamu-A1*056:02:01* (AM295922-Ch) |
| **Mafa-A1*056:03:02** | KT907338 | | |
| *Mafa-A1*064:03* | KT907325 | | |
| *Mafa-A1*065:03* | KT907339 | | |
| *Mafa-A1*065:04:01* | KT907340 | | *Mamu-A1*065:01* (AB430441-Bu, EU418506-Ch) |
| *Mafa-A1*070:01* | KT907341 | ICM[b] | |
| **Mafa-A1*078:03** | KT907344 | | |
| *Mafa-A1*079:02* | KT907342 | ICM[b] | |
| **Mafa-A1*091:02** | KT907319 | | |
| **Mafa-A1*091:03** | KT907320 | | |
| *Mafa-A1*097:01* | KT907318 | ICM[b] | *Mamu-A1*109:01* (AB444902-Bu) |
| **Mafa-A1*099:02** | KT907323 | | |
| *Mafa-A1*130:01* | KT907343 | | *Mane-A1*130:01* (LN875412-Unk), *Mamu-A1*130:01* (HG813262-Unk) |
| *Mafa-A2*01:01* | KT907314 | | *Mamu-A2*01:03* (AB444917-Bu, GQ902066-Ch) |
| *Mafa-A2*05:46* | KT907345 | | *Mamu-A2*05:21* (AM295935-Ch) |
| *Mafa-A3*13:07* | KT907347 | | |

| Allele name | Accession number | Other origin | Macaque counterparts[a] |
|---|---|---|---|
| *Mafa-A4\*14:03* | KT907349 | PCM[b] | *Mamu-A4\*14:03:01* (AB444876-Bu/I, GU080236-Ch) |
| **Mafa-A4\*14:17** | KT907350 | *A4\*14:04* (MaCM)[b] | |

**Notes.**

The 40 *Mafa-A* alleles identified from Vietnamese-origin cynomolgus monkeys are listed. The bold and underlined ones indicate newly identified alleles. IPD name, GenBank accession number, other origin and counterpart(s) in other macaque species are listed for each allele.

[a]For alleles shared with other macaque species, the names of their counterparts, accession numbers, as well as regional populations are also listed. I, Indian rhesus macaque; Bu, Burmese rhesus macaque; Ch, Chinese rhesus macaque; Unk, Unknown-origin rhesus macaque.

[b]For alleles shared identical exons 2 and 3 nucleotide sequences with other populations. ICM, Indonesian origin; PCM, filipino origin; MaCM, Malaysian origin.

[c]For alleles shared identical deduced amino acid sequences encoding $\alpha 1$ and $\alpha 2$ domains with other populations. ICM, Indonesian origin; PCM, filipino origin; MaCM, Malaysian origin.

with 41 of them (17 *Mafa-A*, 24 *Mafa-B*) being identified in our laboratory (*Zhang et al., 2012*; *Zhou et al., 2011*; *Wang et al., 2011*).

Among the 40 *Mafa-A* alleles, 35 sequences originated from the *Mafa-A1* locus, with the other 2, 1, and 2 alleles originating from *Mafa-A2*, *-A3*, and *-A4* loci, respectively. This means that most *Mafa-A* alleles are expressed in the *Mafa-A1* locus (*Pendley et al., 2008*; *Campbell et al., 2009*). As for *Mafa-B* alleles, the locus number designation has not yet been introduced for them because the macaque *MHC-B* genes greatly differ in number between haplotypes (*Shiina et al., 2015*; *De Groot et al., 2012*). Amongst the 30 animals analyzed, each individual expressed 1 to 5 *Mafa-A* genes and 2 to 7 *Mafa-B* genes. On average each monkey expressed 6.8 *Mafa-A/-B* genes. Especially, 11 animals were found to express more than 2 *Mafa-A* sequences. And 14 macaques were found to have more than 4 *Mafa-B* alleles. These data showed that both *MHC-A* and *-B*, especially *-B* genes, were duplicated in cynomolgus macaque of Vietnamese origin. Many of the macaques may contain at least two *Mafa-A* and three *Mafa-B* genes loci. This is similar to the finding of previous articles, which proved that the most frequent *Mafa* haplotype in the Filipino macaque population contains two *MHC-A* and three *MHC-B* loci (*Shiina et al., 2015*; *Kita et al., 2009*).

Among the 100 MHC class I alleles, the most frequently shared *Mafa-A* molecules, containing the same amino acid sequences in exons 2 and 3 with distribution frequency greater than 10%, were *Mafa-A1\*007:01* (8/30, 26.7%), *Mafa-A1\*056:03* (7/30, 23.3%) and *Mafa-A1\*040:03* (4/30, 13.3%). Similarly, the most frequently shared *Mafa-B* alleles with distribution frequency greater than 10% were *Mafa-B\*007:01* (12/30, 40%), *Mafa-B\*039:01* (8/30, 26.7%), *Mafa-B\*060:13* (7/30, 23.3%), *Mafa-B\*093:02* (5/30, 16.7%), *Mafa-B\*101:02* (5/30, 16.7%) , *Mafa-B\*030:17* (4/30, 13.3%), *Mafa-B\*144:01* (4/30, 13.3%) and *Mafa-B\*145:01* (4/30, 13.3%). A summary of the shared alleles and the number of allele clones identified in each macaque are shown in Fig. 1. All of the *Mafa-A1\*007:01*, *Mafa-B\*007:01:01*, *Mafa-B\*039:01* and *Mafa-B\*060:13* were detected to express in individuals 4, 11, and 29, respectively, which indicates that some of them may segregate on one haplotype. Macaque MHC class I allele haplotypes contain variable numbers of loci, which makes them more difficult to characterize than their human counterparts. Although the next-generation sequencing (NGS) techniques have been reported to be effective for high-throughput genotyping of MHC genes and for the detection of low-level-expressed MHC alleles (*Budde et al., 2010*; *Wiseman et al., 2009*), the new technologies are error-prone because it can be more difficult to discriminate between sequencing errors and true

**Table 2  60 *Mafa-B* alleles detected in Vietnamese-origin cynomolgus macaques.**

| Allele name | Accession number | Other origin | Macaque counterparts[a] |
|---|---|---|---|
| *Mafa-B*001:01:01* | KT895485 | | *Mamu-B*001:01:01* (AB477408- Bu, U42837-I) |
| **Mafa-B*006:01:01** | KT895494 | | *Mamu B*006:01* (U41828-Unk) |
| *Mafa-B*007:01:01* | KT895444 | PCM[b] | *Mamu-B*007:03* (AB477412-Bu, EU682528-Ch, AJ556876-I) |
| *Mafa-B*007:01:05* | KT895442 | | |
| *Mafa-B*007:05* | KT895443 | | |
| *Mafa-B*007:08* | KT895446 | | *Mamu-B*007:04:01* (GQ902078-Ch) |
| **Mafa-B*007:09** | KT895445 | | |
| *Mafa-B*013:03* | KT895451 | | |
| *Mafa-B*013:06* | KT895447 | | |
| *Mafa-B*013:09* | KT895448 | *B*013:08* (PCM & ICM)[c] | |
| *Mafa-B*013:10* | KT895449 | | |
| *Mafa-B*013:13* | KT895450 | | |
| *Mafa-B*018:01:01* | KT895490 | ICM[b] | *Mamu-B*018:01* (AM902534-Ch) |
| | | | *Malo-B*018:01* (KT214460-Unk) |
| *Mafa-B*021:02* | KT895452 | | *Mamu-B*021:02* (AM902536-Bu/Ch) |
| *Mafa-B*028:02* | KT895487 | | *Mane-B*028:01* (FJ875264.1-Unk) |
| | | | *Mamu-B*028:02:01* (AM902532.1-Ch) |
| *Mafa-B*028:03* | KT895486 | PCM[b] | |
| *Mafa-B*028:04* | KY131948 | ICM[b] | |
| *Mafa-B*030:01:01* | KT895454 | | *Mamu-B*030:03:02* (AM902546- Ch) |
| *Mafa-B*030:02* | KT895489 | MaCM[b] | *Mamu-B*030:03:03* (AM902547- Ch) |
| *Mafa-B*030:12* | KT895438 | | *Mane-B*030:04* (FJ875259-Unk) |
| *Mafa-B*030:17* | KT895453 | | |
| *Mafa-B*031:01* | KT895491 | | |
| *Mafa-B*034:03* | KT895455 | | |
| **Mafa-B*038:01:02** | KT895456 | | *Mamu-B*038:02* (AB477391-Bu) |
| *Mafa-B*039:01* | KT895457 | | *Maas-B*039:01* (KF012951-Ch) |
| | | | *Mamu-B*039:01* (AB477411-Bu, EF580146-Ch, AJ556890-I) |
| **Mafa-B*039:02** | KT895436 | | |
| **Mafa-B*039:03** | KT895437 | | |
| *Mafa-B*048:04* | KT895458 | | |
| *Mafa-B*050:05* | KT895459 | | |
| **Mafa-B*051:08** | KT895460 | | |
| *Mafa-B*056:01* | KT895488 | ICM[b] | *Mamu-B*056:01* (GQ902079-Ch) |
| *Mafa-B*056:05:01* | KT895461 | | |
| *Mafa-B*060:13* | KT895462 | | |
| *Mafa-B*061:02* | KT895464 | MaCM[b] | |
| *Mafa-B*061:04:01* | KT895463 | | *Mamu-B*061:02* (AM902564-Bu/Ch) |
| *Mafa-B*068:02* | KT895466 | | |

*(continued on next page)*

**Table 2** (*continued*)

| Allele name | Accession number | Other origin | Macaque counterparts[a] |
|---|---|---|---|
| *Mafa-B*068:04* | KT895468 | MaCM[b] | *Mamu-B*068:04* (AM902571-Bu/Ch) |
| *Mafa-B*068:06* | KT895467 | | *Mamu-B*068:02* (EF219482-Unk) |
| **Mafa-B*068:11** | KT895465 | | |
| **Mafa-B*068:12** | KT895469 | *B*068:08* (PCM)[c] | |
| *Mafa-B*069:04* | KT895470 | | |
| *Mafa-B*073:02* | KT895472 | | *Mamu-B*073:01* (AB477404-Bu, AM902578-Ch) |
| **Mafa-B*081:03** | KT895473 | | |
| **Mafa-B*081:04** | KT895474 | *B*081:01* (ICM)[c] | |
| **Mafa-B*082:02** | KT895495 | | |
| *Mafa-B*085:01* | KT895484 | PCM[b] | |
| **Mafa-B*092:02** | KT895471 | *B*092:01* (MaCM)[c] | *Mamu-B*092:02* (AB477386-Bu) |
| *Mafa-B*093:02* | KT895476 | | |
| *Mafa-B*101:02* | KT895493 | | |
| *Mafa-B*104:01:02* | KT895477 | | *Mane-B*104:02* (FJ875231-Unk) |
| *Mafa-B*110:01:01* | KT895478 | | |
| **Mafa-B*112:01** | KT895480 | | |
| *Mafa-B*137:03* | KT895439 | PCM, ICM[b] | |
| **Mafa-B*137:06** | KT895440 | | |
| *Mafa-B*138:02* | KT895479 | MaCM[b] | |
| *Mafa-B*144:01* | KT895482 | | |
| *Mafa-B*145:01* | KT895483 | | |
| **Mafa-B*161:02:02** | KT895481 | | |
| **Mafa-B*180:01** | KT895475 | | |
| **Mafa-B*202:01** | KT895441 | | |

Notes.

The 60 *Mafa-B* alleles identified from Vietnamese-origin cynomolgus monkeys are listed. The bold and underlined ones indicate newly identified alleles. IPD name, GenBank accession number, other origin and counterpart(s) in other macaque species are listed for each allele.

[a]For alleles shared with other macaques species, the name of their counterparts, the accession numbers, as well as regional populations are also listed. I, Indian rhesus macaque; Bu, Burmese rhesus macaque; Ch, Chinese rhesus macaque; Unk, Unknown-origin rhesus macaque.

[b]For alleles shared identical exons 2 and 3 nucleotide sequences with other populations. ICM, Indonesian origin; PCM, filipino origin; MaCM, Malaysian origin.

[c]For alleles shared identical deduced amino acid sequences encoding α1 and α2 domains with other populations, ICM: Indonesian origin. PCM, filipino origin; MaCM, Malaysian origin.

rare alleles. Nevertheless, this problem can be overcome by applying the conventional Sanger sequencing methods. The combined use of the conventional Sanger sequencing methods and the NGS techniques can make the characterization of the highly duplicated macaque *MHC-A/-B* alleles easier to perform (*Shiina et al., 2015*).

**Analysis of alleles shared with other populations or with other species**

The 72 known alleles identified in this study were compared with other populations from Filipino, Indonesian, Malaysian and Mauritian origin. We found that the majority of them (57) were reported previously in Vietnamese origin population (*Zhang et al., 2012*; *Zhou et al., 2011*; *Wang et al., 2011*; *Krebs et al., 2005*), while the remaining 15 alleles share the same exons 2 and 3 sequences with other populations. One of the 15 alleles, namely *Mafa-B*137:03*, was identical to sequences previously described in both Filipino and Indonesian origin populations (*Pendley et al., 2008*; *Shiina et al., 2015*). For the other 14 alleles, 6 of

**Figure 1** Summary of MHC class *I-A* and *-B* alleles identified from 30 cynomolgusmacaques of Vietnamese origin.

them were found identical to Indonesian-origin counterparts (*Pendley et al., 2008*; *Kita et al., 2009*; *Saito et al., 2012*), 4 shared with Filipino-origin population 34and 4 with Malaysian-origin cynomolgus macaque (*Saito et al., 2012*). None allele was found similar to Mauritian-origin population. Interestingly, two sequences of the *Mafa-A4\*14* lineage, *Mafa-A4\*14:03* and *Mafa-A4\*14:17* identified in this study exhibit identical exon 2 and 3 sequences to *Mamu-A4\*14:03*. Meanwhile, the same exon 2 and 3 sequences were shared with Filipino-origin cynomolgus macaque and Malaysian-origin population (*Aarnink et al., 2011*) (Table 1), which indicates that this fragment is conserved in macaque during evolution. Surprisingly, the *Mamu-A4\*14:03* allele was reported to be expressed mainly inside the cell, in contrast to *Mamu-A*-encoded molecules which are mostly found on the cell surface. The different expression patterns were assigned to the antigen-binding $\alpha$1 and $\alpha$2 domains (*Rosner et al., 2010*). It is possible that the two *Mafa-A4\*14* alleles take the same expression pattern in cynomolgus macaque as those for the *Mamu-A4\*14:03* and they have some important functions in the cell rather than presenting peptides on the cell surface to T cells. Meanwhile, there are also 5 alleles possessing the same deduced amino acid sequences encoding $\alpha$1 and $\alpha$2 domains as their counterparts from other populations, including 2 from Malaysia, 1 from Philippines, 1 from Indonesia and the last one shared with Philippines and Indonesia (Tables 1 and 2). No Mauritian origin sequences were matched.

Hence, in this study, we discovered 15 sequences with perfect identity and 6 sequences with identical amino acid sequences encoding $\alpha$1 and $\alpha$2 domains to previously defined MHC class I alleles from Indonesian, Filipino, or Malaysian populations. The sharing of alleles between these geographically distinct populations was consistent with the findings of previous studies, i.e., there is considerable overlap between different populations for some *Mafa-A* or *-B* lineages at the three-digit level of classification, despite the fact that most *Mafa-A* or *-B* alleles are population specific (*Kita et al., 2009*). Therefore, the majority of *Mafa-A* or *-B* alleles in distinct populations probably fine-tuned their sequences to cope with environmental pathogens, along with a few parts inherited conservatively. It is believed that these shared alleles between continental (Vietnamese, Malaysian) and insular (Filipino, Indonesian) subgroups had been generated before the migration of cynomolgus macaques across land bridges between continental Asia and islands of Indonesia during the late Pleistocene epoch (*Kita et al., 2009*).

On the other hand, of the 100 alleles identified, 13 *Mafa-A* and 19 *Mafa-B* sequences were identical to previously reported alleles from other macaque species (Tables 1 and 2). These included the rhesus macaque, the southern pig-tailed macaque (*Macaca nemestrina*, *Mane*), the Northern pig-tailed macaque (*Macaca leonina*, *Malo*) and the Assam Macaque (*Macaca assamensis*, *Maas*). *M.assamensis* inhabits the southern region of Yunnan province, China (*Yan et al., 2013*) and this is the first report of a shared allele (*MHC-B\*039:01*) expressed in cynomolgus, rhesus and assamensis macaques. Another 3 lineages were also shared among at least three macaque species, such as *MHC-A1\*130:01*, *MHC-B\*018:01* and *MHC-B\*028:02*. Interestingly, the shared *MHC-B\*039:01*, was reported in rhesus macaque that it contains a specific B pocket structural motif and has a unique peptide-binding preference consisting of glycine at the second position. This pocket structure was reported

in about 6% of rhesus macaque sequences but absent in human *HLA* genes (*Sette et al., 2012*). Our data showed for the first time that the unique B pocket structural motif also occurred in *Mafa-B*039:01* of Vietnamese origin with high frequency. The biological significance of this molecule needs to be further analyzed in the future and needs to be concerned when using Vietnamese-origin cynomolgus macaques in biomedical researches.

As presented in Tables 1 and 2, five alleles were shared with Indian-origin rhesus macaque. Meanwhile, 19 alleles were shared with Chinese-origin rhesus macaque. It can be easily noticed that cynomolgus macaque shares more alleles with Chinese-origin than with Indian-origin rhesus macaque. This may be explained by the overlap in the geographical areas inhabited by both species in eastern Asia, where the two species are likely to hybridize with extensive ancient introgression from Chinese rhesus macaque into the Vietnamese-origin cynomolgus macaque population, as reported previously (*Bonhomme et al., 2008*; *Stevison & Kohn, 2009*; *Kanthaswamy et al., 2008*). This theory, i.e., ancient hybridization and admixture in macaques, can be also used to explain the fact that the Vietnamese-origin cynomolgus macaque shares more alleles with other macaque species than with other cynomolgus macaque populations (*Fan et al., 2018*). Of these shared alleles between cynomolgus and rhesus macaques, some of them have been found to be associated with diseases in rhesus macaques. For example, *Mamu-B*001* was reported to be resistant to CIA (*Bakker et al., 1992*). Our study showed that the distribution frequency of *Mafa-B*001* is 10% (3/30). It is better to screen out animals containing this allele before building the CIA models with cynomolgus macaques. The other five alleles, *Mafa-A1*001:01*, *Mafa-A1*065:04*, *Mafa-A3*13:02*, *Mafa-B*008:01* and *Mafa-B*017:02* identified in our laboratory (*Bonhomme et al., 2008*; *Kanthaswamy et al., 2008*), are homologous genes of *Mamu-A1*001:01*, *Mamu-A1*065:01*, *Mamu-A3*13:03*, *Mamu-B*008:01* and *Mamu-B*017:01*, which are protective alleles against SIV infection in rhesus macaques. In particular, the distribution frequency of *Mafa-A1*065:04* in this cohort is not low (3/30, 10%). This indicates that cynomolgus macaques with these protective alleles may exhibit delayed AIDS progression and longer survival time after SIV infections. The accumulation of cynomolgus macaques carrying these protective alleles is helpful to analyze the virus-host immune interaction and to gain insights into immune protection against the SIV infection (*Mudd et al., 2012*).

## Analysis of recombination in *Mafa-A* and *Mafa-B* alleles

Recombination event is one of the proposed mechanisms to explain the diversity of MHC alleles. It has been reported that the *Mafa-B*099* allele lineage was generated by the recombination of the *Mafa-B*054* and the *Mafa-B*095* allele lineages in cynomolgus macaque mostly originated from the Philippines (*Orysiuk et al., 2012*). In order to detect the presence of other recombination events in Vietnamese origin cynomolgus macaque, we analyzed 77 *Mafa-A* and 99 *Mafa-B* sequences discovered in our laboratory, including data in this study and those previously reported (*Zhang et al., 2012*; *Zhou et al., 2011*; *Wang et al., 2011*). Using the RDP program, four possible recombination events in *Mafa-A* and *Mafa-B* alleles, as shown in Table S1, were detected by at least four different recombination detection methods. Among them, three recombinants showed lower sequence similarity

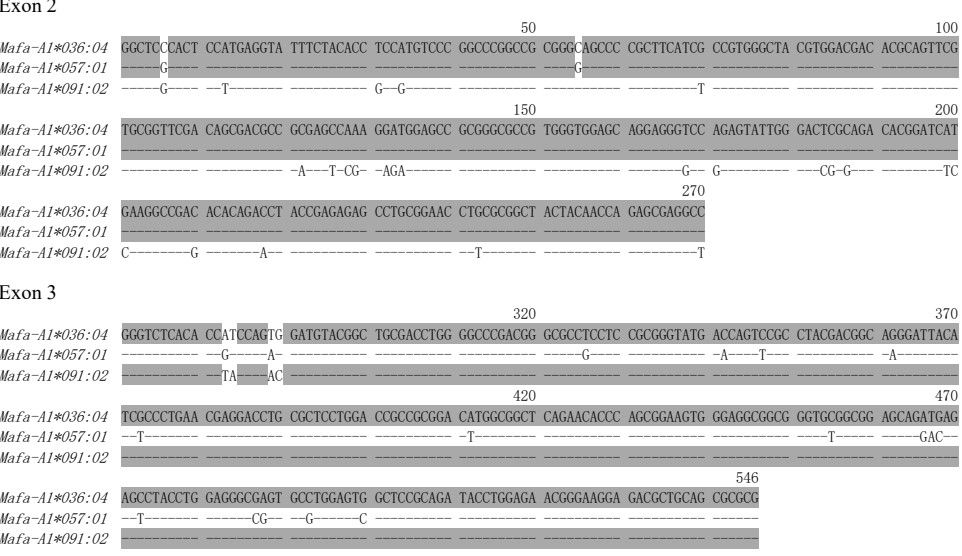

Exon 2

Exon 3

**Figure 2 The nucleotide sequences alignment of *Mafa-A1\*036:04*, *Mafa-A1\*057:01* and *Mafa-A1\*091:02*.**

with their parents hence require further investigation. Only *Mafa-A1\*036:04* was detected as a potential recombinant by five recombination detection methods and exhibited over 98% sequence similarity with the counterparts from its parents. As shown in Fig. 2, the *Mafa-A1\*036:04* and the *Mafa-A1\*091:02* contain very similar sequences in exon 3, with minor variation of four nucleotides in their 5′ regions. However, the exon 2 sequences of these two alleles differ considerably, with 24 nucleotides being mismatched. We also found that the exon 2 sequence of *Mafa-A1\*036:04* is very similar to the *Mafa-A1\*057:01* allele, with only two nucleotide differences present in their 5′ regions. To determine whether the *Mafa-A1\*036:04* was created by a recombination event, we further conducted analysis using RIP (Fig. S1). The result indicates that the *Mafa-A1\*036:04* allele was possibly generated by a crossover event between *Mafa-A1\*091:02* and *Mafa-A1\*057:01*. The exact breakpoint cannot be defined because their intron sequences are not available in this study. Additionally, we further performed phylogenetic analysis of all reported sequences belonging to the *A1\*036*, *A1\*057* and *A1\*091* lineages from cynomolgus and rhesus macaques. The phylogeny map of exon 2 sequences presented in Fig. 3A showed that *Mafa-A1\*036* cluster more closely to Mafa-A1*057 than the *Mafa-A1\*091*. While in the phylogenetic tree of exon 3 presented in Fig. 3B, *Mafa-A1\*036* separates *Mafa-A1\*057* in different branches and groups *Mafa-A1\*091* in a cluster. We also found that the -*A1\*057* and -*A1\*091* lineages are grouped together in these two trees regardless of the species, while the -*A1\*036* lineages are separated according to species. *Mamu-A1\*036* exhibits higher sequence similarity with *Mamu-A1\*091* than with *Mamu-A1\*057*. It is possible that the -*A1\*036* allele generated by crossover recombination between -*A1\*091* and -*A1\*057* just occurred recently in cynomolgus macaque, but not yet in rhesus macaque.

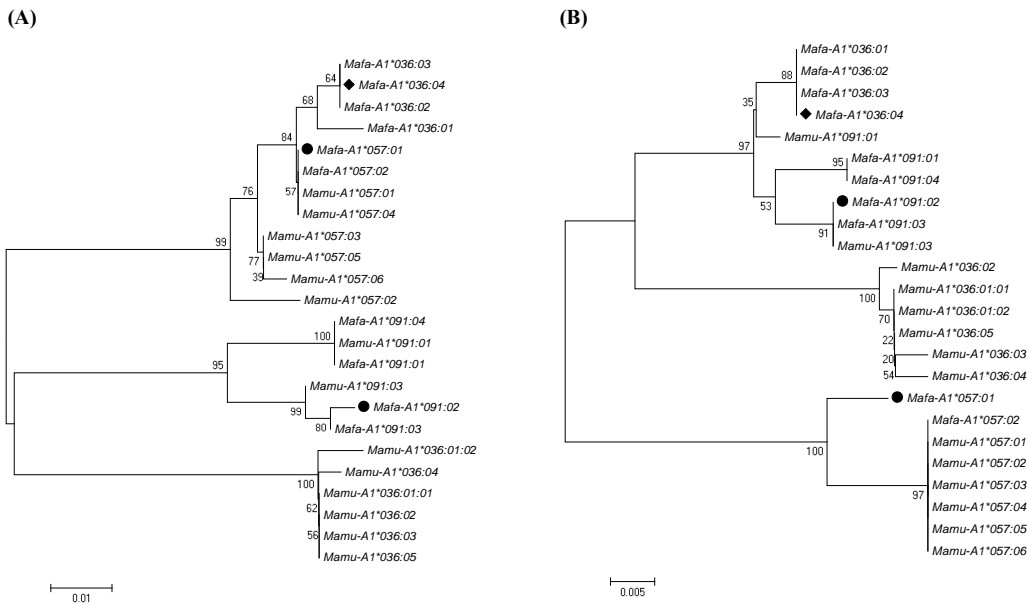

**Figure 3** Phylogenetic analysisof exon 2 (A) and exon 3 (B) of *A1*036,A1*57* and *A1*091* alleles from cynomolgus and rhesusmacaques.

## Analysis of the diversity in *Mafa-A1* locus

We have obtained 77 *Mafa-A* and 99 *Mafa-B* sequences in our laboratory, including 67 *Mafa-A1*. Since *Mafa-A1* was the highest polymorphic gene compared to other *Mafa-A* loci and the locus number designation for *Mafa-B* was not yet clear, here we only analyze the diversity in *Mafa-A1* locus of Vietnamese origin. The sequences of the 67 *Mafa-A1* exons 2 and 3, encoding residues 2–182 of the MHC class I protein, were aligned and a total of 157 nucleotide polymorphic sites (28.8%) were discovered by DnaSP (Table 3). To distinguish positions presenting two or more nucleotides from sites dominated by one nucleotide, we calculated the incidence for the second-most common nucleotide at each position (Table S2). 97 variable sites were considered highly polymorphic with the incidence greater than 5%, while the remaining 60 with the incidence less than 5% were considered to exhibit rare variation. Analysis on the variability index of the second-most common amino acid residue showed that 48 out of 181 sites (26.5%) were defined as highly polymorphic, where the distribution frequency of the second-most common amino acid is greater than 5% (Table S3). In comparison, only 70 nucleotide positions and 45 amino acid residues in *HLA-A* were considered highly polymorphic (*Robinson et al., 2017*). These data showed that cynomolgus macaque *Mafa-A1* exhibit higher polymorphism than human *HLA-A* and several polymorphic sites are macaque-specific (*Kita et al., 2009*). The $\alpha$1 and $\alpha$2 domains of MHC class I glycoproteins contains many functional sites that bind peptide antigens and engage T cell receptors. According to previous studies (*Lafont et al., 2003*), the deduced 36 PBS and 26 TBS were determined. Among these binding sites, 8 residues were both involved in the interaction with the peptides and the receptors. The diversity of nucleotide sequences encoding PBS or TBS ($Pi = 0.162$) in the *Mafa-A1* was predominantly

**Table 3  Polymorphism of exons 2 and 3 sequences for *Mafa-A1* of Vietnamese origin.**

| Site | Ns | S | N | Pi | K |
|------|-----|-----|-----|-------|--------|
| All | 546 | 157 | 218 | 0.073 | 39.886 |
| PBS or TBS | 162 | 81 | 127 | 0.162 | 26.316 |
| Non-PBS or TBS | 384 | 76 | 91 | 0.035 | 13.570 |

**Notes.**

Ns, the number of nucleotides; S, the number of polymorphic sites; N, the number of mutations; Pi, the nucleotide diversity; K, the average number of nucleotide differences.

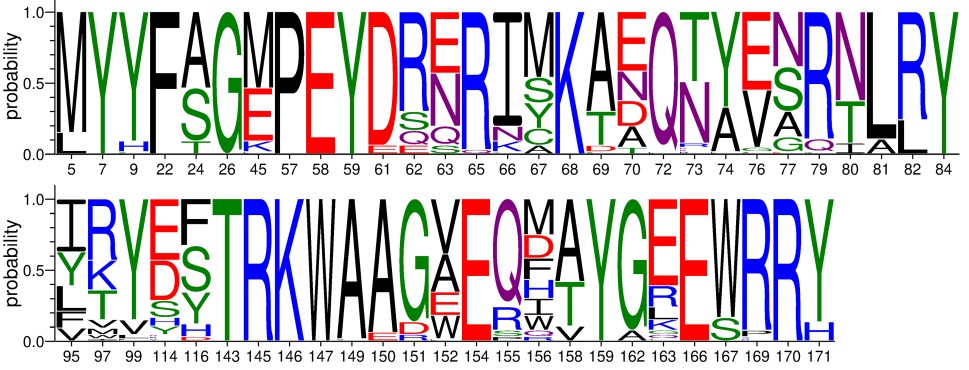

**Figure 4  The diversity of amino acid residues at 54 functional positions relative to PBS or TBS in 67 *Mafa-A1* of Vietnamese origin.**

higher than that in non-PBS or non-TBS coding regions ($Pi = 0.035$) of the corresponding alleles (Table 3). This is consistent with the observation that high polymorphism at these functional residues is significant to increase the depth and breadth of the weaponry to cope with variant pathogens during evolution (*Robinson et al., 2017*; *Lian et al., 2016*). All of these 54 functional residues were listed in Fig. 4, including 34 positions with high polymorphism. On the other hand, 16 out of the 54 functional residues are completely conserved in the 67 *Mafa-A1* sequences. 8 of the 16 residues are also conserved in human, including Y7, Y59, and Y159. The three tyrosine residues were located at an end of the peptide binding groove and may contribute to the recognition of a constant feature of processed antigens (*Bjorkman et al., 1987*) which indicates that these conserved residues are also important to maintain some constant features for presenting peptide and for lymphocyte recognition during evolution.

## CONCLUSION

In this study, we have identified 40 *Mafa-A* and 60 *Mafa-B* alleles from 30 unrelated cynomolgus macaques of Vietnamese origin. 28 of these alleles were found to be novel ones. Each monkey expressed 1 to 5 *Mafa-A* genes and 2–7 *Mafa-B* genes. These data showed that both *MHC-A* and *-B*, especially *-B* genes, were duplicated in cynomolgus macaque of Vietnamese origin. We also identified some alleles with distribution frequency greater than 10% and four alleles (*Mafa-A1*007:01*, *Mafa-B*007:01:01*, *Mafa-B*039:01* and

*Mafa-B*060:13*) were detected to express simultaneously in three individuals. Whether these four alleles segregate on one haplotype need to be verified in future study. Among the 72 known alleles, 15 alleles share the same exons 2 and 3 sequences with other populations, including Filipino, Indonesian and Malaysian origin populations. The sharing of alleles between these geographically distinct populations indicates that a few alleles preserved conservatively in evolution may exercise vital immune functions, and many of the *Mafa-A* or -*B* alleles in distinct populations probably fine-tuned their sequences to cope with environmental pathogens. On the other hand, 32 sequences were identical to previously reported alleles from other macaque species, including 19 shared with Chinese-origin rhesus macaque. The fact that the Vietnamese-origin cynomolgus macaque shares more alleles with Chinese-origin rhesus macaque than with other cynomolgus macaque populations may be explained by ancient hybridization and admixture in macaques. The five alleles identified in our laboratory are homologous genes of protective factors after SIV challenge in rhesus macaque. In this regard, cynomolgus macaque of Vietnamese origin carrying these protective alleles will be a good alternative model to study the immune protection mechanism of SIV infection. To further explain the diversity of *Mafa-A and -B* genes, recombination events and the variability in *Mafa-A1* locus were analyzed, the *Mafa-A1*036:04* allele was possibly generated by a crossover event between *Mafa-A1*091:02* and *Mafa-A1*057:01*, which occurred recently in cynomolgus macaque, but not in rhesus macaque yet. 97 variable nucleotide positions in *Mafa-A1* exons 2 and 3 sequences and 48 amino acid sites in the $\alpha$1 and $\alpha$2 domains were considered highly polymorphic. In comparison to human, the exons 2 and 3 sequences from cynomolgus macaque exhibit higher polymorphism. The information on the diversity of these MHC class I alleles will facilitate the use of Vietnamese-origin cynomolgus macaques to test the therapeutic efficacy and potential side effects of vaccines or other drugs.

## ACKNOWLEDGEMENTS

We are grateful for the Primate Research Center of South China for helping with the cynomolgus macaque blood collection, and for IPD-MHC Nomenclature Committee for providing the designations for the MHC class I alleles.

### Funding

This work was granted by the National Natural Science Foundation of China (No. 31401088) and supported by the Fundamental Research Funds for the Central Universities of South China University of Technology (No. 2014ZZ0056). The funders had no role in study design, data collection and analysis, decision to publish, or preparation of the manuscript.

### Grant Disclosures

The following grant information was disclosed by the authors:

National Natural Science Foundation of China: 31401088.
Fundamental Research Funds for the Central Universities of South China University of Technology: 2014ZZ0056.

## Competing Interests

The authors declare there are no competing interests.

## Author Contributions

- Shuting Huang conceived and designed the experiments, performed the experiments, analyzed the data, prepared figures and/or tables, authored or reviewed drafts of the paper, approved the final draft.
- Xia Huang performed the experiments, analyzed the data, prepared figures and/or tables, approved the final draft.
- Shuang Li and Mingjun Zhu conceived and designed the experiments, contributed reagents/materials/analysis tools, authored or reviewed drafts of the paper, approved the final draft.
- Min Zhuo conceived and designed the experiments, analyzed the data, contributed reagents/materials/analysis tools, prepared figures and/or tables, authored or reviewed drafts of the paper, approved the final draft.

## Field Study Permissions

The following information was supplied relating to field study approvals (i.e., approving body and any reference numbers):

Experiments of using animals and peripheral blood collection were approved by the Institutional Animal Care and Use Committee (Guangdong Landau Biotechnology Co.Ltd.) (project number: IACUC-003).

South China Primate Research & Development Center belongs to Guangdong Landau Biotechnology Co. Ltd. The monkeys we used in experiments were housed in this experimental animal center, and the same monkey feeding and peripheral blood collection were also performed in this experimental animal center. So, the method we used was approved by Landau Biotechnology Co. Ltd.

## DNA Deposition

The following information was supplied regarding the deposition of DNA sequences:

The Mafa-A and Mafa-B sequences are available at GenBank: KT907309–KT907351, KY073130 and KT895436–KT895495, KY131948.

## Data Availability

The raw data are available in the Supplemental Files.

## Supplemental Information

Supplemental information for this article can be found online at http://dx.doi.org/10.7717/peerj.7941#supplemental-information.

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
