# Peer review of "MHC class I allele diversity in cynomolgus macaques of Vietnamese origin"

_PeerJ, doi:10.7717/peerj.7941_

## Round 0.1 · original submission · Major Revisions

Please address all the critiques of the three reviewers and amend your manuscript accordingly.

Reviewer 1 ·

Basic reporting

1. In line 128, a total of 882 Mafa-A and 859 Mafa-B cDNA sequences were obtained, which led to the identification of 100 MHC class I alleles. How many cDNA sequences does each of these alleles have? This information is important because if an allele has many cDNA sequences, it is suggested to be a highly expressed or prevalent allele. In contrast, if an allele has few cDNA sequences, it is suggested to be a lowly expressed or rare allele.
2. In Figure 1, the numbers in the black boxes are not visible.
3. In Figure 2, since Mafa-A1*036:04 is the cross over product of the other two alleles, Mafa-A1*036:04 should be placed at the top row while the other two are in the 2nd and 3rd row for comparison.
4. In Figure 3, figure legend should explain the meaning of the numbers located at the branch points.
5. Line 129, “sequences alignment” should be “sequence alignment”.
6. Line 205 and 317, “to coping with” should be “to cope with”.
7. Line 293, “highly” should be “high”.

Experimental design

1. In order to identify various MHC alleles, the authors produced cDNA from the total RNA, amplified MHC class I genes and performed sequencing. However, these procedures may introduce errors and lead to incorrect identification of novel MHC class I alleles. For example, epitranscriptomic RNA modifications or errors made by reverse transcriptase may introduce false-positive polymorphisms. The authors should discuss how these factors influence the findings of this study.
2. This study found 28 MHC class I alleles, which were not previously identified. What are the improvements in methodology used in this study enable the identification of the novel alleles? For example, are there more animals used in this project to allow more comprehensive characterizations?
3. The authors should further expand the discussion on how findings from this study may contribute to using cynomolgus macaque as a model organism for MHC research.

Validity of the findings

no comment

Additional comments

In this manuscript by Huang et al, the authors studied the diversity of MHC class I genes in cynomolgus macaques of the Vietnamese origin. They identified 28 novel alleles and showed that recombination is one potential source of MHC class I diversity. The manuscript is overall clear and easy to follow. However, the authors need to address the above-mentioned questions.

·

Basic reporting

Overall, the language is fluent and clear, except for some minor grammar mistakes and word usage errors which have been highlighted in the PDF, most of them are plural or singular form related or grammatical tense related mistakes.

On page 7, the author mentioned “the association between MHC I diversity and disease susceptibility ”, we already know that certain MHC I gene in human is associated with certain autoimmune diseases, it would be better if the author could cite some reports of the above association in macaque or other primates.

Experimental design

A couple of previous MHC gene diversity reports of macaques cloned and sequenced the full sequence of MHC I genes, however in this paper, only the exon 2 and 3 are cloned and sequenced, any amino acid mutation in any other domain might change the structure and function of MHC I genes. Please state the reason why clone and sequence the exon 2 and 3 only.

Validity of the findings

For functional immunologic studies, the diversity of amino acid sequence of MHC I genes is more valuable than mRNA sequence, therefore, some readers might be more interested in the amino acid sequence of the novel alleles. It would be more valuable if the author could present the amino acid sequences of novel MHC I alleles in the raw data.

·

Basic reporting

Manuscript is constructed and written in good professional approach, however, it might be better to start the abstract with MHC instead of disease as the reporting is based on MHC alleles. Some typos needs to be fixed like- in title the “cynomolgusmacaques”, abstract line 3rd “tohuman” and elsewhere in the whole manuscript before final submission.

Background information seems sufficient but, at the end of line 66 it would be nice to include a case study like- “Differential Response of the Cynomolgus Macaque Gut Microbiota to Shigella Infection” (https://journals.plos.org/plosone/article?id=10.1371/journal.pone.0064212)

Figures and tables are structured to meet the standards.

Experimental design

The aim of the research by in large fits to the scope of the journal.

The manuscript provides information which would be beneficial to biomedical research, however, more further studies are required for its relevance.

Analysis of recombination in Mafa-A and Mafa-B alleles segment, though the study point is elaborate but would be more convincing if it would include the intron sequences and explain

Validity of the findings

Impact of the study not properly explained with suitable examples, like in conclusion, it would be desired to have more information on the beneficial aspect of shared and unique alleles. Then it would provide readers a direct impact vision in biomedical studies.

In, "Analysis of recombination in Mafa-A and Mafa-B alleles" segment, though the study point is elaborate but would be more convincing if it would include the intron sequences and explain

There are number of speculations in the manuscript but some time its good to provide little rationale at least based on earlier studies like.

Additional comments

The manuscript is good to its original aim, however, it can be more polished with some more information to relate its direct impact on biomedical research and keeping the results analysis tight. Some repetition can be avoided, like- the information on 100 alleles studied and 28 found unique can be constructed more emphatically by discussing on unique ones relevance.

---

## Round 0.2 · accepted · Accept

All the critiques were addressed and the manuscript was revised accordingly. Therefore, this amended version is acceptable now.

Reviewer 1 ·

Basic reporting

The authors have made corresponding changes in text and figures.

Experimental design

The authors have clearly addressed reviewers' questions.

Validity of the findings

The authors have clearly addressed reviewers' questions.

Additional comments

The authors have addressed reviewers' questions and clarified the informations in text/figure.

·

Basic reporting

No comment

Experimental design

No comment

Validity of the findings

no comment

·

Basic reporting

After desired changes, the manuscript seems good in linguistic flow along with scientific approach. There seems to be adequate background context provided and the flow of data matches the context. Hypothesis is justified by the findings and relevant discussion.

Experimental design

Experiments are well designed to the scope of this journal. The findings definitely provides new insight on use of cynomolgus macaques of Vietnamese origin in biomedicine.

Validity of the findings

Data provided are up to the standard of this journal and in sync with the hypothesis. Discussion is little stretchy and could be fitted more. Over all the have come up with good approach to justify their findings.

Additional comments

Authors have successfully remodeled the manuscript story and have made good changes along with complying the concerns raised.